# Percentage of Asymptomatic Infections among SARS-CoV-2 Omicron Variant-Positive Individuals: A Systematic Review and Meta-Analysis

**DOI:** 10.3390/vaccines10071049

**Published:** 2022-06-30

**Authors:** Weijing Shang, Liangyu Kang, Guiying Cao, Yaping Wang, Peng Gao, Jue Liu, Min Liu

**Affiliations:** School of Public Health, Peking University, Beijing 100191, China; shangweijing@bjmu.edu.cn (W.S.); 1610306227@pku.edu.cn (L.K.); caoguiying@bjmu.edu.cn (G.C.); yaping77@bjmu.edu.cn (Y.W.); 1710306136@pku.edu.cn (P.G.); jueliu@bjmu.edu.cn (J.L.)

**Keywords:** COVID-19, SARS-CoV-2, asymptomatic, Omicron variant

## Abstract

Background: Asymptomatic infections are potential sources of transmission for coronavirus disease 2019, especially during the epidemic of the SARS-CoV-2 Omicron variant. We aimed to assess the percentage of asymptomatic infections among SARS-CoV-2 Omicron variant-positive individuals detected by gene sequencing or specific polymerase chain reaction (PCR). Methods: We searched PubMed, EMBASE, and Web of Science from 26 November 2021 to 13 April 2022. This meta-analysis was conducted following the Preferred Reporting Items for Systematic Reviews and Meta-Analyses guidelines and was registered with PROSPERO (CRD42022327894). Three researchers independently extracted data and two researchers assessed quality using pre-specified criteria. The pooled percentage with 95% confidence interval (CI) of asymptomatic infections of SARS-CoV-2 Omicron was estimated using random-effects models. Results: Our meta-analysis included eight eligible studies, covering 7640 Omicron variant-positive individuals with 2190 asymptomatic infections. The pooled percentage of asymptomatic infections was 32.40% (95% CI: 25.30–39.51%) among SARS-CoV-2 Omicron variant-positive individuals, which was higher in the population in developing countries (38.93%; 95% CI: 19.75–58.11%), with vaccine coverage ≥ 80% (35.93%; 95% CI: 25.36–46.51%), with a travel history (40.05%; 95% CI: 7.59–72.51%), community infection (37.97%; 95% CI: 10.07–65.87%), and with a median age < 20 years (43.75%; 95% CI: 38.45–49.05%). Conclusion: In this systematic review and meta-analysis, the pooled percentage of asymptomatic infections was 32.40% among SARS-CoV-2 Omicron variant-positive individuals. The people who were vaccinated, young (median age < 20 years), had a travel history, and were infected outside of a clinical setting (community infection) had higher percentages of asymptomatic infections. Screening is required to prevent clustered epidemics or sustained community transmission caused by asymptomatic infections of Omicron variants, especially for countries and regions that have successfully controlled SARS-CoV-2.

## 1. Introduction

By 11 April 2022, the number of cumulative confirmed coronavirus disease 2019 (COVID-19) cases exceeded 4.9 billion globally, with 6.17 million cumulative deaths [1]. Despite prevention and control measures taken globally to detect and track infected individuals, the transmission induced by asymptomatic individuals remains a dramatic challenge to the global response to COVID-19 [2]. Asymptomatic infected individuals with COVID-19 include pre-symptomatic infection and truly asymptomatic infection, both of which are infectious [3]. Previous studies showed that the viral load in the upper respiratory system of asymptomatic infected persons was similar to that of symptomatic persons and could cause rapid, insidious spread of COVID-19 among the population [4,5,6,7]. Currently, asymptomatic individuals are detected mainly through large-scale population screening and close contact tracing [8]. Temperature screening and symptom monitoring seem to be less effective in pre-symptomatic transmission [3]. Moreover, a previous study also found that a high percentage of asymptomatic infections highlights the potential transmission risk of asymptomatic infections in communities [3].

In November 2021, the first sequenced SARS-CoV-2 Omicron variant case was reported from South Africa [9,10]. Currently, Omicron variant has become the main epidemic strain of COVID-19 worldwide. Etiological studies have shown that mutations in the spike protein cause high infectivity of the SARS-CoV-2 Omicron variant [11,12], and individuals who have been vaccinated and immunized against previous variants (Alpha, Beta, Gamma, Delta) are still susceptible to the Omicron variant [13,14]. In addition, compared with the other variants, the Omicron variant was also characterized by a short incubation period, fast virus transmission, a high percentage of asymptomatic infections, and a low case fatality rate [11,15,16]. All the characteristics have undoubtedly hindered the prevention and control of the epidemic. From November 2021 to January 2022, the Omicron variant was associated with travel history in South Africa and spread swiftly globally, which made cluster and community transmission increase significantly in many countries [6,7,17,18]. Children, adolescents, and adults are susceptible to the Omicron variant, and some show no symptoms at the initial stage of infection [15,19]. Therefore, asymptomatic infection identification in communities and schools is critical for Omicron prevention and advance preparation.

Two countries have reported publicly available data of asymptomatic infections, including a community-wide national representative surveillance program in England [20,21] and electronic reporting of diagnostic laboratory test results, TestCenter Denmark, in Denmark [19,22]. However, the percentage of asymptomatic infections among SARS-CoV-2 Omicron variant-positive individuals is unclear. Current results vary considerably due to different study designs, populations, and regions. Cross-sectional studies have shown that the percentage of asymptomatic Omicron-infected individuals ranged from 7.9% to 61.0% [23,24,25,26,27,28,29,30] during the epidemic period of the Omicron variant, while this percentage in cohort studies ranged from 1.23% to 43.7% [31,32,33,34]. Thus, we conducted a meta-analysis to better understand the percentage of asymptomatic infections among SARS-CoV-2 Omicron variant-positive individuals detected by gene sequencing or specific polymerase chain reaction (PCR). Our results could be useful for the estimation of potentially infected individuals, priority population screening, and policy making.

## 2. Materials and Methods

### 2.1. Search Strategy

We conducted this meta-analysis following the Preferred Reporting Items for Systematic Reviews and Meta-Analyses (PRISMA) guidelines. This review was registered with PROSPERO (CRD42022327894). Three researchers (W.S., L.K., and G.C.) searched the published studies between November 26, 2021 and April 13, 2022, through PubMed, EMBASE, and Web of Science with English-language restriction. The search terms included (“SARS-CoV-2” or “COVID-19” or “Omicron”) and “asymptomatic infections”. The detailed search strategies are shown in Appendix A. Three researchers (W.S., L.K., and G.C.) independently reviewed the titles, abstracts, and full texts of articles and identified additional studies from the reference lists. Disagreements were resolved by 2 other reviewers (Y.W. and P.G.).

In this study, SARS-CoV-2 Omicron variant-positive individuals were defined as persons with a confirmed sequencing result for SARS-CoV-2 Omicron or an S-gene target failure on a specific PCR assay [9,35]. The asymptomatic infected persons referred to those who did not present any symptoms at the time of SARS-CoV-2 Omicron variant testing or diagnosis [3].

### 2.2. Inclusion and Exclusion Criteria

The inclusion criteria consisted of: (1) studies reporting the number of SARS-CoV-2 Omicron variant-positive individuals and asymptomatic Omicron-positive individuals, and (2) cross-sectional studies or cohort studies. Exclusion criteria consisted of: (1) articles unable to find full text; (2) reviews, letters, and guidelines; (3) detection time before Omicron variant; (4) articles unable to extract data; (5) multiple studies reporting on overlapping participants (the study with more information was included); and (6) studies with less than 100 SARS-CoV-2 Omicron variant-positive individuals.

### 2.3. Data Extraction

Three researchers (W.S., L.K., and G.C.) performed the data extraction independently. Data were extracted for the first author, publication year, study location, study design, number of SARS-CoV-2 Omicron variant-positive individuals, and number of asymptomatic infected individuals. The transmission place, travel history (yes or no), vaccine coverage (proportion of positive individuals who received one or more doses of vaccine), ratio of male to female individuals (MFR), and median age of Omicron variant-positive individuals were gathered if available.

### 2.4. Risk of Bias Assessment

The quality of the studies included in the meta-analysis was assessed using the Joanna Briggs Institute Prevalence Critical Appraisal Tool [3,36] for cross-sectional studies and the Newcastle–Ottawa scale [37] for cohort studies (Appendix A). Two researchers (W.S. and L.K.) independently performed the quality assessment. Disagreements were resolved by 2 other reviewers (G.C. and Y.W.). The outcome of interest was the percentage of asymptomatic infections among the SARS-CoV-2 Omicron variant-positive individuals.

### 2.5. Data Synthesis and Statistical Analysis

We performed a meta-analysis to estimate the pooled percentage of asymptomatic infections among SARS-CoV-2 Omicron variant-positive individuals. Untransformed percentages and DerSimonian and Laird random-effects models were used to calculate the pooled percentage and its 95% confidence interval (CI) [38]. The heterogeneity among studies was assessed using *I*^2^ values, and very low, low, moderate, and high degrees of heterogeneity were defined as ≤25%, 25% to ≤50%, 50% to ≤75%, and ≥75%, respectively [39]. We performed subgroup analyses by country development level (developed vs. developing), publication year (2021 vs. 2022), sample size of the Omicron variant-positive individuals (<500 vs. ≥500), study design (cohort studies vs. cross-sectional studies), study quality (low and moderate), transmission place (community vs. others), travel history (no vs. yes), vaccine coverage (<80% vs. ≥80%), MFR (0.5 to <1.0 vs. 1.0 to 1.5), and median age (<20 vs. 20–40 years). We performed 2 sensitivity analyses to test the robustness of our results by using Knapp–Hartung adjustments [40] to calculate the 95% CIs around the pooled effects and by excluding studies with low quality. Two-sided *p* < 0.05 indicated statistical significance. All analyses were performed using R, version 4.0.5 (R Project for Statistical Computing).

## 3. Results

### 3.1. Characteristics of Included Studies

We identified 6553 studies through a database search and the reference lists of articles and reviews. Of these, 541 studies underwent full-text review. Eight studies [19,21,32,33,34,41,42,43] with information concerning the percentage of asymptomatic infections among SARS-CoV-2 Omicron variant-positive individuals were included in the final analysis (Figure 1). The characteristics of the studies included in this systematic review and meta-analysis are shown in Table 1.

Among these studies, three (37.5%) were cohort studies, and five (62.5%) were cross-sectional studies. Four studies (50.0%) were conducted in Europe, two (25.0%) in Asia, one (12.5%) in North America, and one (12.5%) in Africa. Five studies (62.5%) were conducted in developed countries. Seven studies (87.5%) were published in 2022. Five studies (62.5%) had a sample size ≥ 500. Five studies (62.5%) were assessed as moderate quality, and three (37.5%) were assessed as low quality. Three studies (37.5%) and two studies (25.0%) were relevant to community transmission and travel history, respectively.

### 3.2. Percentage of Asymptomatic Infections among the SARS-CoV-2 Omicron Variant-Positive Individuals

Eight studies, a total of 7640 Omicron variant-positive individuals, were included in the meta-analysis to estimate the percentage of asymptomatic infections. Among them, 2190 had asymptomatic infections, with the pooled percentage of asymptomatic infections of 32.40% (95% CI: 25.30–39.51%). Heterogeneity among studies was high (*I*^2^ = 97.7%; *p* < 0.001) (Figure 2).

Figure 3 shows the results of the subgroup analysis. The pooled percentage of asymptomatic infections was higher in developing countries (38.93%; 95% CI: 19.75–58.11%) than in developed countries (28.66%; 95% CI: 20.69–36.63%). The pooled percentage was 33.72% (95% CI: 25.55–41.89%) in studies published in 2022, and 34.12% (95% CI: 22.21–46.03%) in cohort studies. The pooled percentage was higher in studies relevant to community transmission (37.97%; 95% CI: 10.07–65.87%) and travel history (40.05%; 95% CI: 7.59–72.51%). The studies with vaccine coverage ≥ 80% (35.93%; 95% CI: 25.36–46.51%) had a higher pooled prevalence than those with coverage < 80% (31.23%; 95% CI: 12.83–49.62%). Among studies with MFR of 1.0 to 1.5, the pooled percentage was higher (40.05%; 95% CI: 7.59–72.51%). The pooled percentage was higher when the median age of asymptomatic infected persons was <20 years (43.75%; 95% CI: 38.45–49.05%).

### 3.3. Publication Bias and Sensitivity Analysis

Since there are fewer than 10 studies in the meta-analysis, which is not enough to use tests for funnel plot asymmetry [44,45], we did not draw a conclusion on publication bias. After using the Knapp–Hartung adjustments, the pooled percentage of asymptomatic infections was 32.40% (95% CI: 20.99–43.82%) among the SARS-CoV-2 Omicron variant-positive individuals, and the 95% CI of the pooled percentage became slightly larger (Appendix A). After excluding three low-quality studies, the pooled percentage of asymptomatic infections among the SARS-CoV-2 Omicron variant-positive individuals was 37.48% (95% CI: 27.79–47.16%), slightly higher than the original results.

## 4. Discussion

In this meta-analysis, we found that the pooled percentage of asymptomatic infections was 32.40% (95% CI: 25.30–39.51%) in SARS-CoV-2 Omicron variant-positive individuals detected by gene sequencing or specific PCR, and there was potential heterogeneity across studies. We found that the percentage of asymptomatic infections was higher in developing countries and in infected persons with a vaccination coverage rate ≥ 80%, travel history, community transmission, and median age < 20 years. In the sensitivity analyses, we found that the pooled percentage was higher than the original results after excluding low-quality studies.

In this study, the pooled percentage of asymptomatic infections was 32.40% among the SARS-CoV-2 Omicron variant-positive individuals. At present, there is no systematic review reporting the percentage of asymptomatic infections among SARS-CoV-2 Omicron variant-positive individuals detected by gene sequencing or specific PCR. Although the related studies were limited due to the short epidemic time of the Omicron variant, our systematic review suggests that there are still a substantial number of asymptomatic infected individuals who may cause potential transmission. Additionally, the percentage of asymptomatic infections among SARS-CoV-2 Omicron variant-positive individuals still needs to be further explored by future studies.

In this study, the percentages of asymptomatic infections among SARS-CoV-2 Omicron variant-positive individuals detected by gene sequencing or specific PCR were different among developing and developed countries. Studies conducted in developing countries had the highest percentage of 38.93%. South Africa implemented large-scale population screening and active hospital surveillance after the initial report of Omicron variant. However, this action may lead to a higher percentage of asymptomatic infections [46]. In addition, the study conducted in a population with a vaccination coverage rate ≥ 80% also had a higher asymptomatic infections percentage of 35.93%. Most individuals infected with COVID-19 before are likely to be reinfected due to the increased immune escape capability of the Omicron variant, which may cause a higher percentage of asymptomatic infections [46,47]. Previous studies showed that the Omicron variant-positive individuals with booster doses of vaccines had a lower percentage of severe symptoms and a higher percentage of no symptoms or mild symptoms [48,49]. This suggests that the current vaccine is still effective in preventing severe cases but seems to be less effective in preventing Omicron infections. Updated vaccines are needed to provide better protection.

In our study, the pooled percentage of asymptomatic infections was 40.05% and 37.97% among the Omicron variant-positive individuals having a travel history and infected in the community, respectively. This finding suggested that screening and quarantine of asymptomatic infected travelers [19,43,50] are important to prevent community transmission. Routine screening of community residents, especially of those commuting frequently [3], may prevent some cluster epidemics or community transmission. In addition, we found that the pooled percentage of asymptomatic infections was 43.75% in Omicron variant-positive individuals with the median age < 20 years. Insufficient vaccine coverage due to vaccine worries or late vaccination of first and booster, and clustering characteristics of this population (mainly students) may lead to a high percentage of asymptomatic infections [3,34]. In this meta-analysis, the median age of Omicron variant-positive individuals in all eight studies was no more than 40 years old, and most of them showed no symptoms initially when infected because of high baseline immunity [15,51]. This indicates that this group has potential for transmission of the Omicron variant, and routine screening may prevent the occurrence of clustered epidemics in communities, schools, and other places.

In this study, we included studies published from 26 November 2021 to 13 April 2022, which can provide the most updated pooled percentage of asymptomatic infections among the SARS-CoV-2 Omicron variant-positive individuals. To our knowledge, this is the first study that investigated the percentage of asymptomatic infections among Omicron variant-positive individuals detected by gene sequencing or specific PCR. We also estimated the percentage of asymptomatic infections in different vaccination coverage rates, age groups, transmission places, and in populations having travel history. Our results could raise awareness among the public and policy makers and provide evidence for prevention strategies.

This study has several limitations. First, the Omicron variant appeared within in a short time; thus, only a few related studies were available, which limits the number of studies in our systematic review. Second, the heterogeneity between studies was high, which might be related to different locations, populations, and sample sizes. Therefore, it was still necessary to monitor and update our systematic review when new publications or data meeting our criteria became available.

## 5. Conclusions

In this systematic review and meta-analysis, we found that the pooled percentage of asymptomatic infections was 32.40% among SARS-CoV-2 Omicron variant-positive individuals detected by gene sequencing or specific PCR. The people who were vaccinated, young (median age < 20 years), had a travel history, and were infected outside of a clinical setting (community infection) had a higher percentage of asymptomatic infections. Screening is required to prevent clustered epidemic or sustained community transmission caused by asymptomatic infections of Omicron variants, especially for countries and regions that have successfully controlled SARS-CoV-2.

## Figures and Tables

**Figure 1 vaccines-10-01049-f001:**
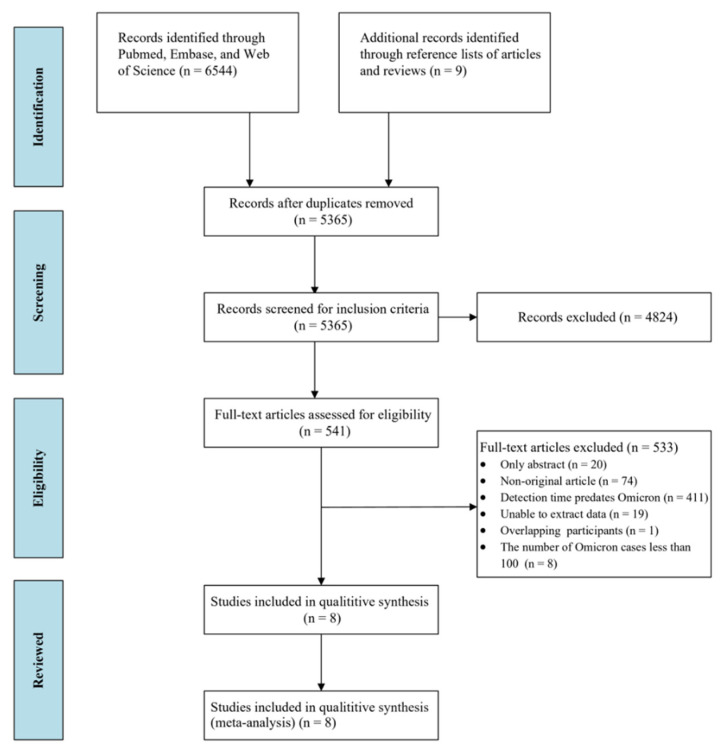
Flowchart of the study selection.

**Figure 2 vaccines-10-01049-f002:**
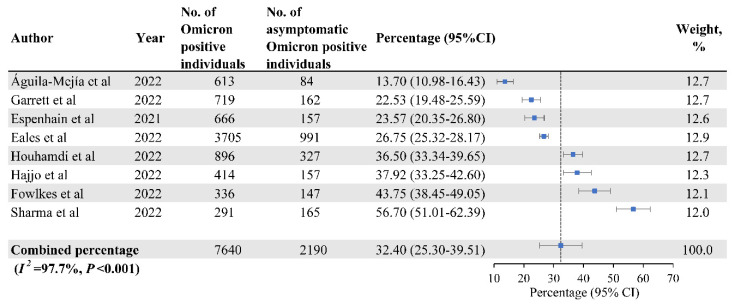
The Percentage of Asymptomatic Infections Among SARS-CoV-2 Omicron Variant-Positive Individuals [19,21,32,33,34,41,42,43].

**Figure 3 vaccines-10-01049-f003:**
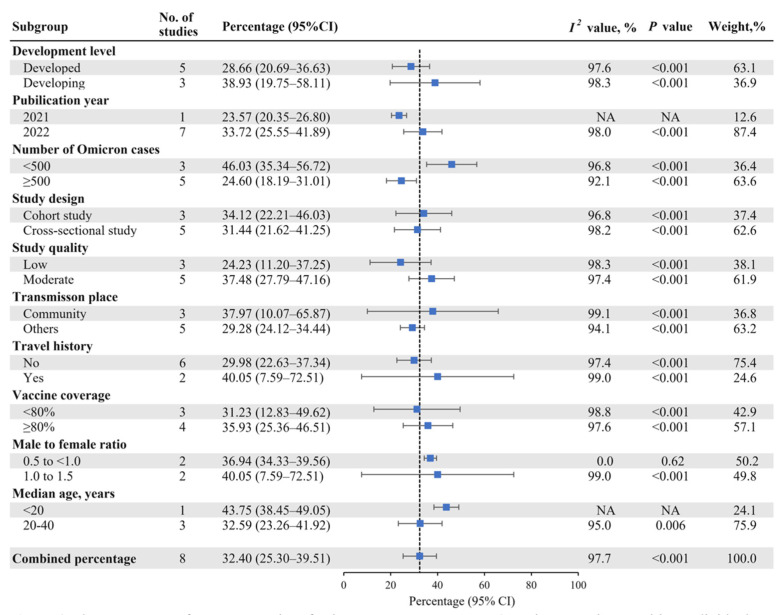
The Percentage of Asymptomatic Infections Among SARS-CoV-2 Omicron Variant-Positive Individuals by Subgroup.

**Table 1 vaccines-10-01049-t001:** Characteristics of the Studies Included in the Systematic Review and Meta-analysis.

**First Author, Year**	**Country**	**Study Design**	**No. Omicron** **- Positive Individuals, n**	**No. Asymptomatic Omicron Positive Individuals, n**	**MFR**	**Median Age, y**
Águila-Mejía et al. (2022) [41]	Spain	Cross-sectional	613	84	/	/
Eales et al. (2022) [21]	UK	Cross-sectional	3705	991	/	/
Espenhain et al. (2021) [19]	Denmark	Cross-sectional	666	157	1.23	32
Fowlkes et al. (2022) [34]	US	Cohort	336	147	/	<20
Garrett et al. (2022) [32]	South Africa	Cohort	719	162	/	/
Hajjo et al. (2022) [42]	Jordan	Cross-sectional	414	157	0.98	30
Houhamdi et al. (2022) [33]	France	Cohort	896	327	0.83	33
Sharma et al. (2022) [43]	India	Cross-sectional	291	165	1.31	/
**First Author, Year**	**Country Development Level**	**Travel History**	**Transmission Place**	**Vaccine Coverage,%**	**Study Quality**	**PMIDs**
Águila-Mejía et al. (2022) [41]	Developed	No	Community	73.41	Low	35393009
Eales et al. (2022) [21]	Developed	No	Other	89.80	Moderate	/
Espenhain et al. (2021) [19]	Developed	Yes	Other	85.50	Moderate	34915977
Fowlkes et al. (2022) [34]	Developed	No	Community	55.36	Moderate	35298453
Garrett et al. (2022) [32]	Developing	No	Other	/	Low	35353885
Hajjo et al. (2022) [42]	Developing	No	Other	91.70	Moderate	/
Houhamdi et al. (2022) [33]	Developed	No	Other	46.37	Low	35060146
Sharma et al. (2022) [43]	Developing	Yes	Community	86.20	Moderate	/

## Data Availability

Data can be obtained by contacting the corresponding author.

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
