# Peer review of "Percentage of Asymptomatic Infections among SARS-CoV-2 Omicron Variant-Positive Individuals: A Systematic Review and Meta-Analysis"

_vaccines, 2022, doi:10.3390/vaccines10071049_

Round 1

Reviewer 1 Report

In the introduction, consider adding a review of any publicly available databases/resources that provide this same type of information outside of publications. If not, maybe in the discussion, consider adding that public databases/resources might be a good source of additional data to be added to your dataset.

Table 1, it would be nice to add the PMIDs for each paper and the vaccination rates of the countries, to get an idea about the community environment of the subjects

In Discussion, consider adding if you will continue to monitor if new publications meeting your criteria are published and if are you planning an update after x new publications/data points become available. 

grammar/style comments:

line 24 reword "infection in communities" to "community infection" because the term "communities" implies a specific community, such as the immunosuppressed, the inner city, the rural, etc. and I think you mean infection outside of a clinical setting? if so, then the phrase  "community infection"  is more appropriate.

line 29: similarly, consider rewording "infected in communities" to "infected outside of a clinical setting (community infection)"

throughout: consistently insert a space before all brackets used for citations.

line 47: insert "to be" between "seem" and "less", replace "this" with "a"

line 60: insert "was" after "variant", add "and" before "spread"

line 62: insert "the" before "Omicron"

line 63: remove "of them"

lines 64-65: completely reword

line 74:insert "the estimation of" after "for", change "individual" to "individuals" and remove the word "estimation"

Figure 1: typos in the word "synythesis", should be "synthesis"

line 221: add "the" before "community"

line 222: change "on" to "of"

line 223: change both uses of "on" to "of"

line 232: insert "for" before "transmission"

line 244: insert "within" before "in"

line"245: change "are" to "were"

Author Response

Response to Reviewer 1 Comments

We gratefully appreciate for your valuable comments.

Point 1: In the introduction, consider adding a review of any publicly available databases/resources that provide this same type of information outside of publications. If not, maybe in the discussion, consider adding that public databases/resources might be a good source of additional data to be added to your dataset.

Response 1: Thanks for the Reviewer’s suggestion. Considering the Reviewer’s suggestion, we have added a review of two publicly available databases/resources that provide asymptomatic infections information outside of publications. The details were shown in Line 66 to 69 in the manuscript. Detailed descriptions: Two countries have reported publicly available data of asymptomatic infections, including community-wide national representative surveillance program in England and electronic reporting of diagnostic laboratory test results, TestCenter Denmark, in Denmark.

Point 2: Table 1, it would be nice to add the PMIDs for each paper and the vaccination rates of the countries, to get an idea about the community environment of the subjects.

Response 2: Thank you for your suggestion. In Table1, we have added the PMIDs for each paper and the vaccination rates of the countries. Detailed descriptions were shown in page 5, line 157-158.

Point 3: In Discussion, consider adding if you will continue to monitor if new publications meeting your criteria are published and if are you planning an update after x new publications/data points become available. 

Response 3: Thanks for the Reviewer’s suggestion. In the discussion, we have considered and added that we would monitor and update our systematic review when new publications or data meeting our criteria available. Detailed descriptions were shown in page 8, line 252-254.

Point 4: grammar/style comments:

line 24 reword "infection in communities" to "community infection" because the term "communities" implies a specific community, such as the immunosuppressed, the inner city, the rural, etc. and I think you mean infection outside of a clinical setting? if so, then the phrase "community infection" is more appropriate.

Response 4: Thanks for your suggestion. We reworded "infection in communities" to "community infection" in page 1 line 24 and checked the whole paper to avoid the same "mistake".

Point 5: line 29: similarly, consider rewording "infected in communities" to "infected outside of a clinical setting (community infection)"

Response 5: Thanks for your suggestion. We reworded "infected in communities" to " infected outside of a clinical setting (community infection)" in page 1 line 28-29 and checked the whole paper to avoid the same "mistake".

Point 6: throughout: consistently insert a space before all brackets used for citations.

Response 6: Thanks for your suggestion. We inserted a space before all brackets used for citations in whole paper.

Point 7: line 47: insert "to be" between "seem" and "less", replace "this" with "a"

Response 7: Thank you for your suggestion. We inserted "to be" between "seem" and "less" and replace "this" with "a" in page 2 line 47.

Point 8: line 60: insert "was" after "variant", add "and" before "spread"

Response 8: Thanks for your suggestion. We inserted "was" after "variant", add "and" before "spread" in page 2 line 60.

Point 9: line 62: insert "the" before "Omicron"

Response 9: Thanks for your suggestion. We inserted "the" before "Omicron" in page 2 line 62.

Point 10: line 63: remove "of them"

Response 10: Thank you for your suggestion. We removed "of them" in page 2 line 63.

Point 11: lines 64-65: completely reword

Response 11: Thanks for your suggestion. We completely reworded the sentence in page 2 line 63-65. Details were shown below.

“Therefore, asymptomatic infections identification in communities and schools is critical for Omicron prevention and advance preparation.”

Point 12: line 74: insert "the estimation of" after "for", change "individual" to "individuals" and remove the word "estimation"

Response 12: Thanks for your suggestion. We inserted "the estimation of" after "for", change "individual" to "individuals" and removed the word "estimation" in page 2 line 77-78.

Point 13: Figure 1: typos in the word "synythesis", should be "synthesis"

Response 13: Thanks for your suggestion. We changed the word "synythesis" to "synthesis" in page 4 figure1.

Point 14: line 221: add "the" before "community"

Response 14: Thanks for your suggestion. We added "the" before "community" in page 7 line 227.

Point 15: line 222: change "on" to "of"

Response 15: Thanks for your suggestion. We changed "on" to "of" in page 7 line 227.

Point 16: line 223: change both uses of "on" to "of"

Response 16: Thanks for your suggestion. We changed both uses of "on" to "of " in page 7 line 229.

Point 17: line 232: insert "for" before "transmission"

Response 17: Thanks for your suggestion. We inserted "for" before "transmission" in page 7 line 238.

Point 18: line 244: insert "within" before "in"

Response 18: Thanks for your suggestion. We inserted "within" before "in" in page 8 line 249.

Point 19: line"245: change "are" to "were"

Response 19: Thanks for your suggestion. We changed "are" to "were" in page 8 line 250.

Reviewer 2 Report

In this meta-analysis, authors have assessed the percentage of asymptomatic infections among SARS-CoV-2 Omicron variant positive individuals. The meta-analysis is important considering the rise of COVID-19 infections worldwide and asymptomatic infections plays a major role in the continuous wave of infections.

There are few concerns which needs to be rectified

·      Figure 2 and 3 table is blurred, needs to be submitted in better resolution

·      Global word in the title and manuscript needs to be changed as only 7, 640 Omicron variant positive individuals were included in the meta-analysis.

·      Is there any specific reason why the percentage of asymptomatic infections was higher in developing countries?

·      There are spelling mistakes and grammatical error which needs to be corrected

Author Response

Response to Reviewer 2 Comments

We gratefully appreciate for your valuable comments.

Point 1: Figure 2 and 3 table is blurred, needs to be submitted in better resolution.

Response 1: Thank you for your reminding. I have submitted figure 2 and 3 table in better resolution.

Point 2: Global word in the title and manuscript needs to be changed as only 7, 640 Omicron variant positive individuals were included in the meta-analysis.

Response 2: Thanks for your suggestion. I have deleted the word “global” in the title and manuscript.

Point 3: Is there any specific reason why the percentage of asymptomatic infections was higher in developing countries?

Response 3: Thanks for your suggestion. The potential reasons are as follows. First, we think that the high vaccination coverage rate in the article included is the possible specific reason for higher percentage of asymptomatic infections in developing countries. According to our results of the subgroup analysis, the studies with vaccine coverage ≥ 80% (35.93%; 95% CI: 25.36%-46.51%) had a higher pooled percentage of asymptomatic infections than those with coverage < 80% (31.23%; 95% CI: 12.83%-49.62%). Moreover, previous studies also reported that vaccination population have higher rate of Omicron asymptomatic infections. In the studies included in our meta-analysis, two of three studies conducted in developing countries have higher vaccination coverage rate (≥80%), while vaccination coverage was lower (<80%) in three of five studies in developed countries. Additionally, infected individuals with mild symptoms might be diagnosed as confirmed cases due to higher monitoring, detection and diagnosis technology in developed countries. In the future, more researches about asymptomatic infection of the Omicron variant are needed in developed countries and continue expanding vaccine coverage in developing countries. We have already mentioned main reasons above in the discussion section.

Point 4: There are spelling mistakes and grammatical error which needs to be corrected

Response 4: Thank you for your suggestion. The manuscript has also been double-checked and spelling mistakes and grammatical errors we found have been corrected.
